# First Digits’ Shannon Entropy

**DOI:** 10.3390/e24101413

**Published:** 2022-10-03

**Authors:** Welf Alfred Kreiner

**Affiliations:** Faculty of Natural Sciences, University of Ulm, Einsteinallee 11, D-89069 Ulm, Germany; welf.kreiner@uni-ulm.de

**Keywords:** alphabet, Boltzmann, calcite, Clausius, craters, density function, entropy, exoplanets, first-digit phenomenon, fragment, granite, information, marble, mineral, Newcomb–Benford law, populations, probability, scale invariance, Shannon entropy, space probes, statistical thermodynamics, stock prices, street addresses, Venus, wages

## Abstract

Related to the letters of an alphabet, entropy means the average number of binary digits required for the transmission of one character. Checking tables of statistical data, one finds that, in the first position of the numbers, the digits 1 to 9 occur with different frequencies. Correspondingly, from these probabilities, a value for the Shannon entropy H can be determined as well. Although in many cases, the Newcomb–Benford Law applies, distributions have been found where the 1 in the first position occurs up to more than 40 times as frequently as the 9. In this case, the probability of the occurrence of a particular first digit can be derived from a power function with a negative exponent *p* > 1. While the entropy of the first digits following an NB distribution amounts to H = 2.88, for other data distributions (diameters of craters on Venus or the weight of fragments of crushed minerals), entropy values of 2.76 and 2.04 bits per digit have been found.

## 1. Introduction

According to information theory, the Shannon entropy H of a variable is the average level of information inherent to the variable’s possible outcomes [1]. Originally, the term entropy was coined by Clausius [2] for a state variable in thermodynamics. Boltzmann [3] related the entropy of an ideal gas to the multiplicity, the number of microstates corresponding to the gas´s macrostate. From the probability of occurrence of letters, Shannon calculated the entropy H of an alphabet [4,5]. This gives the minimum number of binary digits necessary for the transmission of one character and, simultaneously, means its average information. In quite a similar way, one can obtain a value of H from the probability of occurrence of the first digits in tables of numbers. This gives the average information of a first digit and the number of binary digits required for their transmission as well. Due to Kwiatkowsky [6], entropy is a measure of uncertainty.

One goal of this manuscript is to apply Shannon´s method to first digits of data collections. The entropy of the first digits, giving the minimum of binary digits necessary to characterize their distribution, may be of some value in case the signal to noise ratio is extremely low, e.g., when transmitting data from space probes. 

Another goal is to find first digits´ distributions that follow a continuous function, such as the data that obey Newcomb–Benford´s Law, but on the other hand exhibit a clearly different frequency of the first digits. Examples seem to be rare.

The question is also discussed regarding distributions of the first digits that are very different from the NB Law. In the case of fractured minerals, a possible explanation is presented.

For an alphabet, the entropy H is determined in the following way [7]: if the letters occur with different probabilities, *p*_i_, one forms the logarithmus dualis of a particular character, ld *p*_i_, and multiplies this value with the probability of its occurrence in the text, giving *p*_i_ ld *p*_i_. Taking the sum over all letters results in
(1)$$H=-\overset{n}{\underset{i=1}{\sum }}\left(p_{i}\;\mathrm{ld}\;p_{i}\;\right)\;$$
with *n* being the total number of letters [8,9] in this alphabet. The average information of a single character shows its highest value when all letters are equally likely. Correspondingly, one can calculate a value for the entropy H from the probability of occurrence of first digits in data collections as well. 

## 2. The Newcomb–Benford Distribution 

Checking tables of statistical data (e.g., atomic weights, masses of exoplanets, but also wages and prices, stock prices, populations of communities, street addresses, lengths of rivers, page numbers in literature citations, development aid), one finds quite often that 1 occurs more frequently in the first position of numbers than 9. The best-known example is the Newcomb–Benford distribution, saying that the number 1 occurs 6.58 times more often as the leading digit than 9 [10,11,12,13]. Some continuous processes satisfy this exactly (in the asymptotic limit as the process continues through time). Examples are exponential growth or decay processes [12]: if a quantity is exponentially increasing or decreasing in time, then the percentage of time that it has each first digit satisfies Newcomb–Benford’s Law. An example is given in Appendix A. Many—though not all—distributions follow the NB Law more or less closely (the population of municipalities, microorganisms in a culture, spread of internet contents, capital growth, wages and prices, loss of value over time, street addresses [12], page numbers in literature citations [14]). Biau [15] analyses the discrepancies between the NB Law and first-digits frequencies in data of turbulent flows through Shannon’s entropy.

For a more general approach to the problem of first digits´ probabilities, one starts from the density function
(2)$$D(x)\;\sim \;\frac{1}{x^{P}}={\;x}^{-P}$$

The probability W(a,b) of the object to exhibit a size between a and b then is
(3)$$W\;(a,b)=W(z)=A\int_{a}^{b}x^{-P}dx\;$$

In case of exponential growth of a certain object, x means the time and *p* equals 1.
(4)$$W\;(a,b)=W(z)=A\int_{a}^{b}x^{-1}dx=A\;ln\frac{b}{a}$$

The bars in Figure 1 refer to subsequent integers a, b, defining a “z-interval“.

Equations (3) and (4) are scale invariant, i.e., one would obtain the same power law and the same ratio of 1s to 9s in the first digital place if one would use other units to measure the objects´ properties, as long as one chooses comparable intervals (e.g., one decade).

### 2.1. Entropy of the Newcomb–Benford Distribution

Similar to the letters of an alphabet, the probability of occurrence can be determined from the first digits of a data set following the Newcomb–Benford Law. In Table 1, the values for W(z), 1 ≤ z ≤9, are listed, together with the probability of occurrence of a particular digit in the first position of numbers. From the probabilities, one finds the numerical value of the entropy with Equation (1) of the first digits´ distribution to be H (NB Law) = 2.8762 bits per digit. This means the average information of a single first digit. In case the leading digits were evenly distributed (all p_i_ = $\frac{1}{9}$), the entropy would amount to H = −$9\;\frac{1}{9}\;ld\left[\frac{1}{9}\right]\;$= 3.1699 bpd.

### 2.2. Non-NB Distributions 

In case the first digits´ distribution does not follow the NB Law (Figure 2), it can be derived from the density function D(x) = $\frac{A}{x^{P}}={\;\mathrm{Ax}}^{-P}$ with *p* ≠ 1. The proportion of numbers falling into the range between x_1_ and x_2_ is calculated from
(5)$$W(z)=W(x_{1},\;x_{2})=A\int_{x_{1}}^{x_{2}}D\left(x\right)dx=A\int_{x_{1}}^{x_{2}}x^{-P}\mathrm{dx}=\frac{A}{1-P}\;\left[x_{2}{}^{1-P}-{\;x}_{1}{}^{1-P}\right]$$

This law is scale invariant. In case x_1_ and x_2_ are chosen as subsequent integers, W(x_1_, x_2_) = W(z) is in proportion to the numbers starting with the digit x_1_ = z. From the probabilities of the first digits, the average entropy of a digit can be determined (Equation (1)).

## 3. Mass Distributions of Minerals

### 3.1. Fragments of Marble

The distribution of the first digits of fractions of minerals occasionally deviates strongly from the NB Law [16]. Six samples of marble (irregularly shaped plates of 10 mm thickness, weighing between 4.70 and 51.69 g) were crushed in a hydraulic press, and the weights of 1052 fragments between 10 and 99 mg were taken. Fragments below 10 and above 99 mg have been omitted. In Table 2, the numbers of fragments sharing the same first digit are listed. From the probabilities, p_i_, the value of the entropy H = 2.0370 has been found. 

From a fit of Equation (5) (red curve in Figure 3), the value of *p* = 2.003 (25) has been determined. Assuming the distribution would follow exactly this function, the statistical weights and the first digits´ probabilities can be obtained from
(6)$$W\;(a,b)=W(z)=A\int_{a}^{b}x^{-P}\mathrm{dx}=\frac{A}{-1.003}[b^{-1.003}-a^{-1.003}\;]$$
where a, b are the integers defining the z intervals. 

In this case, a 1 (a = 1, b = 2) is expected to occur as a leading digit 45.27 times more often than a 9 (a = 9, b = 10). For statistical reasons, the observed ratio is different (58.8). It is, e.g., closer for the ratio of the 1 s to the 7 s (calculated: 29.8, observed: 30.9).

#### Why Is the Number 1 So Strong in The Majority of Cases?

It is surprising that, when it comes to the weight of the fragments, the number 1 occurs even more often as a leading digit than expected from the NB Law. To answer this question, one has to determine the probability of the weight of a fragment exhibiting a certain first digit.

Table 3 shows the number of first digits for this case, so one would divide the sample into the number of equal-sized fragments given in the left column. For example, cutting a mass of 81,475 g (a weight chosen arbitrarily) into 815 pieces of equal size, one finds that the weight of each piece starts with a 9. This holds up to 905 fragments. Dividing the sample into smaller parts (between 906 and 1018 fragments) results in an 8 in the first position. Between 4074 and 8147 fragments of equal size, the leading digit will be a 1. Although the sample will probably never break into pieces of equal size, there will be at least a high probability for a majority of the fragments that their weight will start with a 1. A 9 will be rare. Table 4 gives the probabilities of occurrence of the first digits.

From Table 5, one can see that, in this example, the ratio of 1 s to 9 s is 45265/1006 = 44.995. Starting from samples with different weights produces quite similar results. 

Figure 4 gives the distribution of probabilities among the z-intervals. From a fit (Equation (5); Figure 4), one obtains *p* = 2.0 for the exponent of the density function. From this value, the ratio of the 9 s to the 1 s (occurring as a first digit) should be exactly 45. From the experiments, the ratios 42.12 (calcite), and 27.62 (granite) have been calculated (Section 3.2 and Section 3.3).

Table 3 also shows that there is kind of periodicity in the occurrence of a certain first digit when dividing the sample into more and more pieces. Inspecting the table from top to bottom, one finds that the number of a particular digit increases approximately by a factor of 10. 

### 3.2. Fragments of Calcite

From 2 samples of calcite, 775 fragments between 10 and 99 mg were obtained. Table 6 and Figure 5 give the distribution with respect to the first digits of their weight. A value for the entropy of H = 2.1064 is obtained. The exponent of the density function amounts to *p* = 1.966 (36).

From the function, the probabilities for W(z = 1) and W(z = 9) have been calculated. Their ratio is 42.12. From the counted number of fragments, a ratio of 423/7 = 60.43 is obtained.

### 3.3. Fragments of Granite

Samples of granite (13 square discs, 34 × 34 × 9.5 mm^3^, 28 g each, with 3 more samples, slightly larger) were crushed in a hydraulic press. The weight of 3849 fragments between 10 mg and 99 mg was taken. The distribution is given in Figure 6 as well as in Table 7. From this, a value of H = 2.1817 bits per digit has been found. 

From a fit of Equation (5) (red curve in Figure 6), the value for the parameters *p* = 1.748 (81) has been determined. Assuming the distribution would follow exactly this function, the statistical weights and the first digits´ probabilities can be obtained from
(7)$$W\;(a,b)=W(z)=A\int_{a}^{b}x^{-P}\mathrm{dx}=\frac{A}{-0.748}[b^{-0.748}-a^{-0.748}\;]$$

From this, a 1 is expected to occur as a leading digit 27.62 times more often than a 9. From the counted number of fragments, the ratio is found to be larger (1923/51 = 37.71).

The distribution of the first digits would be close to the same power law when fragments are selected in the comparable 10-fold range of 1.0 × 10^−4^ ounces to 9.9 × 10^−4^ ounces.

## 4. Prices of Shares and Equity Funds

Equation (5) can be used to obtain a *p*-value from share prices. On a certain day in 2022, a listing of 397 share prices was taken (including stock indices and funds; Table 8 exhibits the distribution of first digits given in Figure 7 (prices in €)). From the frequency of the first digits, the entropy has been determined to H = 2.841 bpd. A fit of Equation (5) resulted in *p* = 1.045 (74). Usually, for share prices, the distribution of the first digits deviates only slightly from the NB Law [17]. A 1 occurs as a first digit 7.17 times as frequently as a 9 (from the function fitted).

## 5. Craters on the Venus

The first digits of the diameters of 874 craters on Venus [18] produced an H value of 2.764 bits per digit (Table 9). Figure 8 shows the distribution. Due to the function fitted (*p* = 1.238(73)), a 1 should be 10.36 as frequent as a 9.

## 6. Results

The entropy of most of the first digits´ distributions in listings encountered in daily life exhibits a value around H = 2.9 bits per digit, while in some cases (diameters of craters on the planet Venus, weights of mineral fragments), considerably lower values have been found. Table 10 gives the results.

The largest deviation of the entropy from the NB distribution was observed for the weight of fragments of some minerals. Whether this is due to material constants or whether it can be attributed to the experimental conditions has to be left to future investigations.

## Figures and Tables

**Figure 1 entropy-24-01413-f001:**
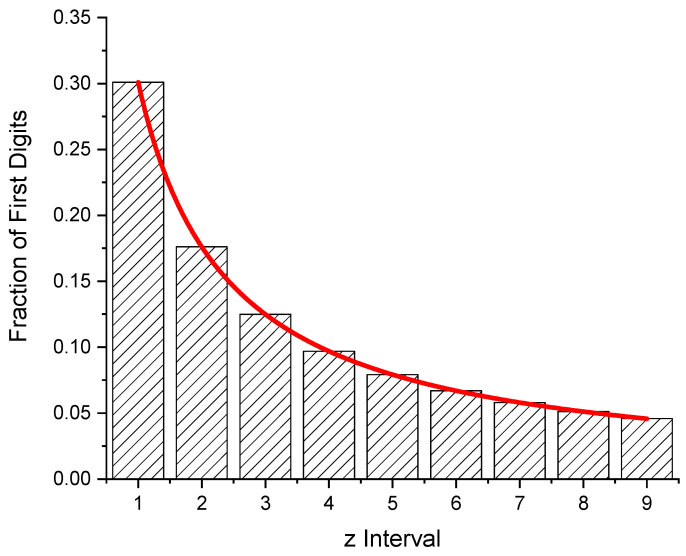
The Newcomb–Benford probability distribution of first digits. The probability of a 1 being the leading digit is 6.58 times as high as a 9. The red curve shows W(z), Equation (4).

**Figure 2 entropy-24-01413-f002:**
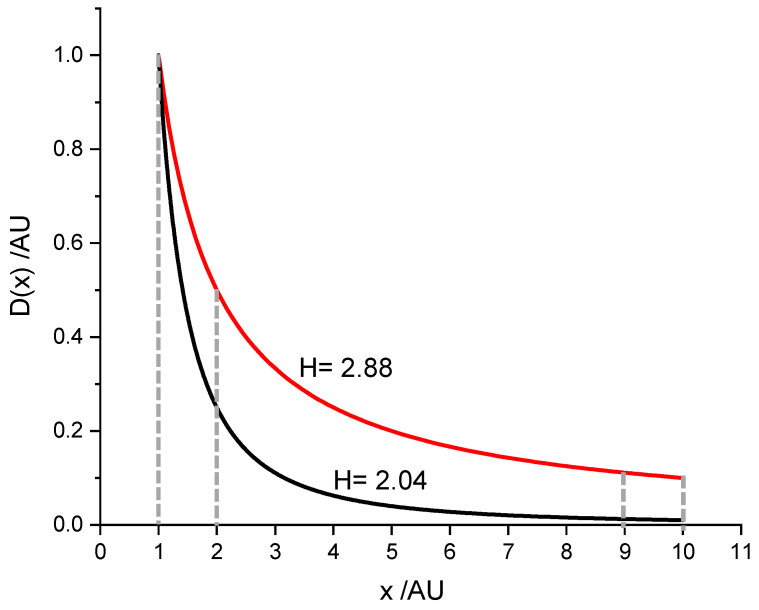
Density function of a first digit distribution obeying the NB Law (D(x) ~ x^−1^; red) and of a distribution deviating from it ((D(x) ~ x^−2.003^; black). The areas between the dashed lines correspond to the amount of data starting with a 1 or a 9, respectively. The black curve gives the density function of the first digit distribution of the weight of samples of crushed marble, as shown in Figure 3.

**Figure 3 entropy-24-01413-f003:**
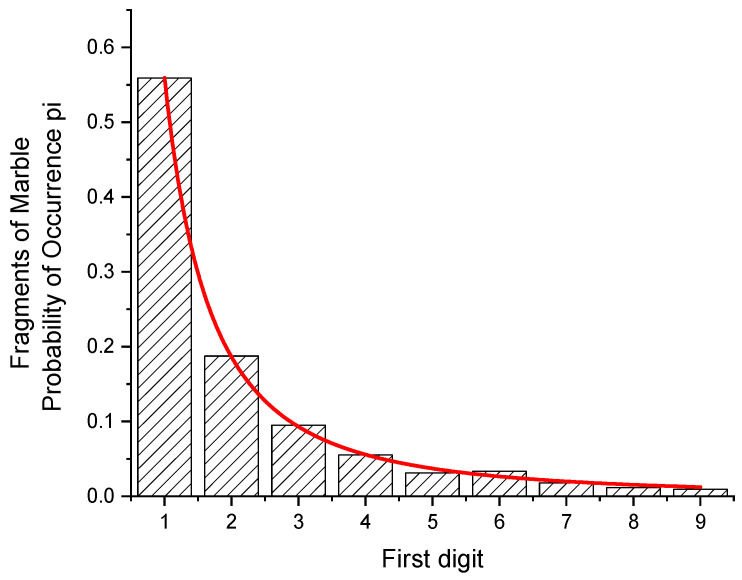
Fragments of six samples of marble. Fit of the function W(z) to a bar graph giving the probabilities of first digits of the weights of 1052 fragments between 10 and 99 mg. From the probabilities of occurrence of the first digits, a value of the entropy H = 2.0370 bits per digit has been found. The function fitted corresponds to a density function with *p* = 2.003 (25).

**Figure 4 entropy-24-01413-f004:**
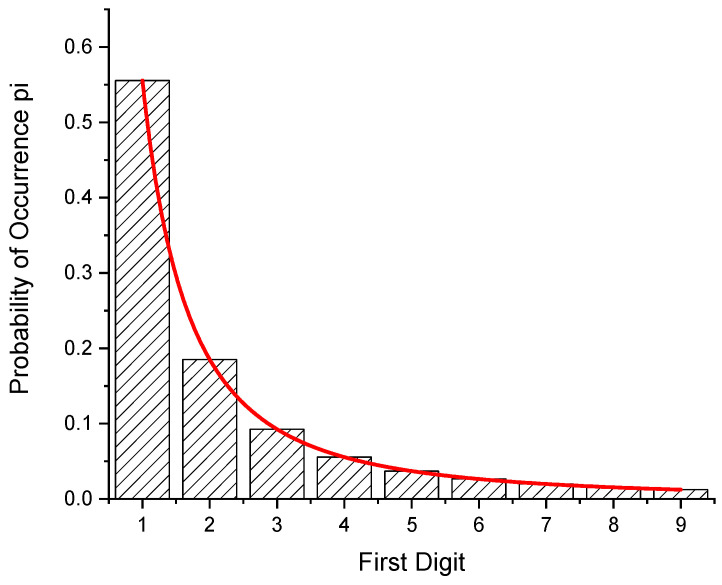
Probability of first digits when the sample is divided hypothetically into fragments of equal size, as shown in Table 3. From the fit of the probability function, W(z), for the density function exponent, a value of *p* = 2.00003 (4) is obtained.

**Figure 5 entropy-24-01413-f005:**
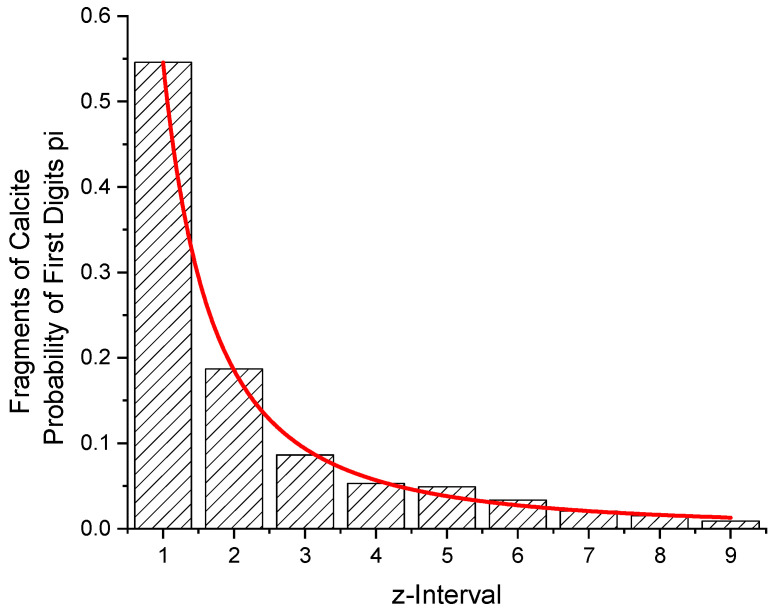
Fit of the function W(z) to a bar graph giving the probabilities of first digits of the weight of 775 fragments of calcite between 10 and 99 mg. *p* = 1.966 (36). From the function, the probabilities for W(z = 1) and W(z = 9) have been calculated. Their ratio amounts to 42.12.

**Figure 6 entropy-24-01413-f006:**
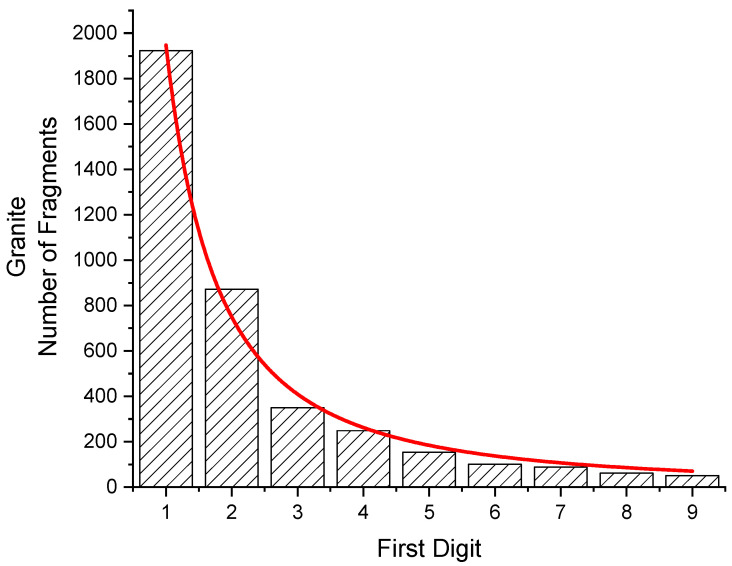
Fit of the function W(z) to a bar graph giving the probabilities of first digits of the weight of 3849 fragments of granite between 10 and 99 mg (obtained from 15 samples). *P* = 1.748 (81). From the function, the probabilities for W(z = 1) and W(z = 9) have been calculated. Their ratio is 27.62. From the counted number of fragments, the ratio is 37.71.

**Figure 7 entropy-24-01413-f007:**
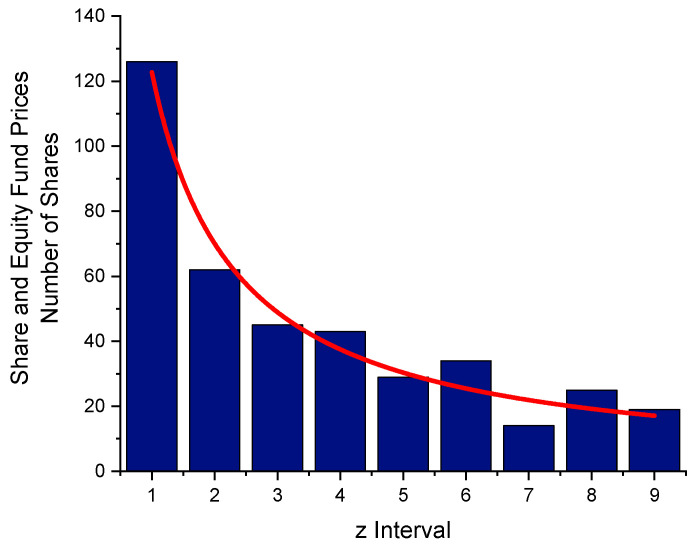
Share and equity fund prices. From the first digits´ distribution of 397 stock prices for H, a value of 2.841 bits per digit has been derived. From a fit, *p* = 1.045(74) has been found. This agrees with the NB Law within its error limit.

**Figure 8 entropy-24-01413-f008:**
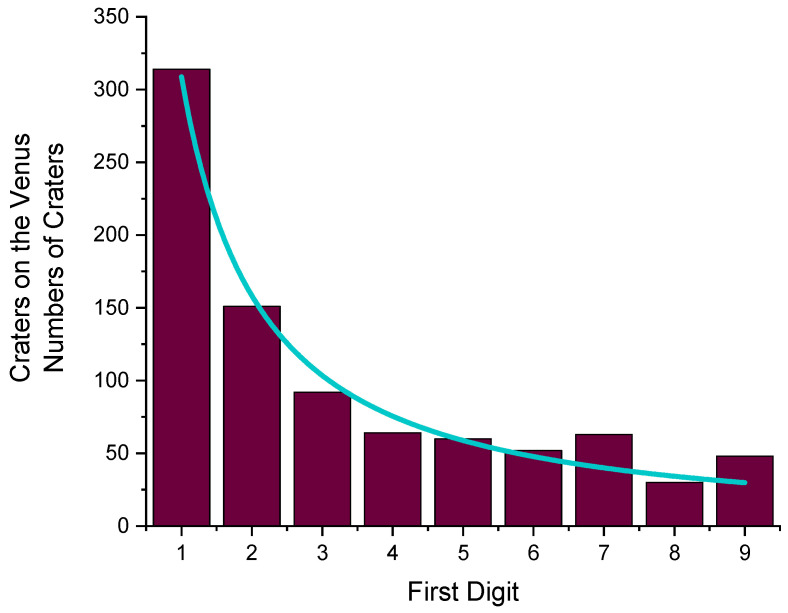
Distribution of the first digits of the diameters of 874 craters on Venus. H = 2.764 bits per digit. From a fit (blue curve), *p* = 1.238 (73) has been derived.

**Table 1 entropy-24-01413-t001:** Entropy H of the first digits´ distribution due to the Newcomb–Benford law.

a = z	$W(a,b)=ln\frac{b}{a}$	b	p_i_	−p_i_ ld p_i_
1	0.6931	2	0.3010	0.5214
2	0.4055	3	0.1761	0.4412
3	0.2877	4	0.1249	0.3748
4	0.2231	5	0.0969	0.3263
5	0.1823	6	0.0792	0.2897
6	0.1542	7	0.0670	0.2613
7	0.1335	8	0.0580	0.2383
8	0.1178	9	0.0512	0.2195
9	0.1054	10	0.0458	0.2037
**∑**	**2.3026**		**1.0001**	**H = 2.8762** bits per digit

**Table 2 entropy-24-01413-t002:** Probabilities of fragments of marble.

z Interval	Number of Fragments	p_i_	−p_i_ ld p_i_
1	588	0.5589	0.4691
2	197	0.1873	0.4526
3	100	0.0951	0.3227
4	58	0.0551	0.2305
5	33	0.0314	0.1567
6	35	0.0333	0.1633
7	19	0.0181	0.1046
8	12	0.0114	0.0736
9	10	0.0095	0.0638
**∑**	1052	1.0001	**H = 2.0370** bits per digit

**Table 3 entropy-24-01413-t003:** Number of fragments of a mass of 81,475 g with identical first digits.

Number of Fragments	1	2	3	4	5	6	7	8	9
9									1
10								1	
11							1		
12–13						2			
14–16					3				
17–20				4					
21–27			8						
28–40		13							
41–81	42								
82–90									9
91–101								11	
102–116							15		
117–135						19			
136–162					27				
163–203				41					
204–271			68						
272–407		136							
408–814	407								
815–905									91
906–1018								113	
1019–1163							145		
1164–1357						194			
1358–1629					272				
1630–2036				407					
2037–2715			679						
2716–4073		1358							
4074–8147	4074								
8148–9052									905
9053–10,184								1132	
10,185–11,639							1455		
11,640–13,579						1940			
13,580–16,295					2716				
16,296–20,368				4073					
20,369–27,158			6790						
20,159–40,737		13,579							
40,738–81,475	40,738								
**∑**	**45,265**	**15,088**	**7545**	**4526**	**3018**	**2155**	**1616**	**1257**	**1006**

**Table 4 entropy-24-01413-t004:** Probabilities of occurrence of first digits.

	1	2	3	4	5	6	7	8	9
**p_i_**	**0.55556**	**0.18518**	**0.09260**	**0.05555**	**0.03704**	**0.02645**	**0.01983**	**0.01543**	**0.01235**

**Table 5 entropy-24-01413-t005:** Ratio 1 vs. 9 and entropy H.

Ratio 1 vs. 9: 45265/1006 = 44.995 Entropy H = $-\overset{9}{\underset{1}{\sum }}p_{i}ld\left(p_{i}\right)=$ 2.0692

**Table 6 entropy-24-01413-t006:** First digits´ entropy of samples of calcite.

z Interval	Number of Fragments	p_i_	−p_i_ ld p_i_
1	423	0.5458	0.4768
2	145	0.1871	0.4524
3	67	0.0865	0.3053
4	41	0.0529	0.2243
5	38	0.0490	0.2133
6	26	0.0336	0.1643
7	16	0.0207	0.1256
8	12	0.0155	0.0931
9	7	0.0090	0.0613
**∑**	775	1.0001	**H = 2.1064** bits per digit

**Table 7 entropy-24-01413-t007:** First digits´ entropy of samples of granite.

z Interval	Number of Fragments	p_i_	−p_i_ ld p_i_
1	1923	0.4996	0.5002
2	872	0.2266	0.4853
3	350	0.0909	0.3145
4	248	0.0644	0.2549
5	154	0.0400	0.1858
6	102	0.0262	0.1378
7	88	0.0229	0.1246
8	62	0.0161	0.0959
9	51	0.0133	0.0827
**∑**	3849	1.0000	**H = 2.1817** bits per digit

**Table 8 entropy-24-01413-t008:** First digits´ distribution of the prices of 397 funds (in €).

**First Digit**	1	2	3	4	5	6	7	8	9	∑
**Frequency**	126	62	45	43	29	34	14	25	19	397
**p_i_**	0.3174	0.1562	0.1134	0.1083	0.0730	0.0856	0.0353	0.630	0.0479	1.0001
**−p_i_ (ld p_i_)**	0.523	0.407	0.338	0.309	0.325	0.278	0.148	0.219	0.294	
							**H = 2.841** bits per digit			

**Table 9 entropy-24-01413-t009:** First digits´ distribution of the diameters of 874 craters on Venus.

**First Digit**	1	2	3	4	5	6	7	8	9	∑
**Frequency**	314	151	92	64	60	52	63	30	48	874
**p_i_**	0.3593	0.1728	0.1053	0.0732	0.0686	0.0595	0.0721	0.0343	0.0549	1.0000
**−p_i_ (ld p_i_)**	0.5306	0.4376	0.3420	0.2761	0.2652	0.2422	0.2735	0.1669	0.2299	
								**H = 2.7640** bits per digit		

**Table 10 entropy-24-01413-t010:** Values for first digit entropy for different distributions.

Data Context	H/Entropy Bits per Digit	Exponent *p* of the Density Power Function	Ratio of First Digits (1 vs. 9)
Even distribution of first digits	**3.170**	**0**	**1**
Newcomb–Benford distribution	**2.876**	**1**	**6.58**
Share prices (from the density function fitted)	**2.841**	**1.045 (74)**	**7.17**
Diameters of 874 craters on Venus (from the density function fitted)	**2.764**	**1.238 (73)**	**10.36**
Weights of pieces of granite (from the density function fitted)	**2.182**	**1.748 (81)**	**27.62**
Weights of pieces of calcite (from the density function fitted)	**2.106**	**1.966 (36)**	**42.12**
Weights of pieces of marble (from the density function fitted)	**2.037**	**2.003 (25)**	**45.27**

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
