# Peer review of "First Digits’ Shannon Entropy"

_entropy, 2022, doi:10.3390/e24101413_

Round 1

Reviewer 1 Report

I am puzzled by this paper. On the one hand there is no novelty at all, and all results are entirely obvious. On the other, the study is quite interesting and reading the paper is a nice experience.

The overall conclusion of the presented study seems to be that the first digit, in practical situations where it follows the Newcomb-Benford law, has entropy H \approx 2.9 bits. This is of course obvious from the assumption about the distribution but the paper also verifies the conclusion through several numerical examples. For some reason there is a special emphasis on minerals.

Overall I think the readers of this journal would find the paper a nice read with some interesting observations. However, since there is very little scientific novelty I lean toward not recommending the paper for publication.

Author Response

Reviewer1:

“On the one hand there is no novelty at all, and all results are entirely obvious.“

A major reason for this publication is to point out that there are distributions, where the frequencies of the first digits follow a continuously decreasing function like the data obeying the NB Law, but on the other hand, exhibit a distribution quite different from NB.

“For some reason there is a special emphasis on minerals.“:

Fractured minerals provide a rare example of a non-NB distribution. This is all the more remarkable as another mass distribution, that of 845 exoplanets, follows the NB law nearly exactly.

As the practical purpose of calculating a value of the first digits´ entropy one can imagine that in case of extremely low signal to noise ratio it may be of some value to get knowledge at least about the first digits´ distribution of measured data. The entropy H provides this information via a minimum number of bits to be transferred.

Reviewer 2 Report

The writing style shows German influence --- the decimal point is sometimes portrayed with a , instead of the more familiar . .  But there is inconsistency in this --- sometimes within a line of separation both forms are used.  Also, lower and upper quotations ,,quote'' are used, as opposed to the more familiar ''quote''.  These should be made consistent with more conventional English usage.

There is some confusion about units on entropy in the introduction, and lack of precision in statements.  This should be clarified.  In particular, the question of whether the digits (or characters) are independent needs to be clarified.  If the first digit has a preponderant value of 1, is the 2nd digit  independent of it? 

The paper is interesting, but I am not sure that I perceive any "value" in the sense that it will have relevance to better scientific understanding of measured data.  Some discussion of "why this matters" would be helpful.

I have uploaded the document with several comments.

Author Response

Reviewer 2

Thank you for your comments. Decimal points have been set instead of commas where needed, the quotation mark has been corrected.

“There is some confusion about units on entropy in the introduction, and lack of precision in statements.“       

The sentence referring to Boltzmann has been replaced.

“What sort of tables do you mean?“

Not only scientific data (like atomic weights, masses of exoplanets), but also wages and prices, stock prices, populations of communities, street addresses, length of rivers, page numbers in literature citations, development aid.

Round 2

Reviewer 1 Report

I maintain that this work shows very little scientific novelty. There might still be an interest among potential readers though, and I will not object to accepting the paper.

Author Response

Thank you very much for your comments.

Kind regards, WA Kreiner

Reviewer 2 Report

The authors have addressed concerns and provided appropriate clarifications.

Author Response

(The authors gave the same response as above.)
